# Biodegradable Nanoparticle for Cornea Drug Delivery: Focus Review

**DOI:** 10.3390/pharmaceutics12121232

**Published:** 2020-12-18

**Authors:** Mohammadmahdi Mobaraki, Madjid Soltani, Samaneh Zare Harofte, Elham L. Zoudani, Roshanak Daliri, Mohamadreza Aghamirsalim, Kaamran Raahemifar

**Affiliations:** 1Department of Biomedical Engineering, Amirkabir University of Technology, Tehran 15875‐4413, Iran; m.mahdimobaraki70@gmail.com; 2Translational Ophthalmology Research Center, Tehran University of Medical Science, Tehran 1417614411, Iran; aghamirsalim@gmail.com; 3Department of Electrical and Computer Engineering, Faculty of Engineering, University of Waterloo, Waterloo, ON N2L 3G1, Canada; 4Faculty of Science, School of Optometry and Vision Science, University of Waterloo, Waterloo, ON N2L 3G1, Canada; kraahemi@ryerson.ca; 5Department of Mechanical Engineering, K. N. Toosi University of Technology, Tehran 19967-15433, Iran; samanehzare@gmail.com (S.Z.H.); Elham.lzn@yahoo.com (E.L.Z.); Roshanakdaliri@gmail.com (R.D.); 6Centre for Biotechnology and Bioengineering (CBB), University of Waterloo, Waterloo, ON N2L 3G1, Canada; 7Advanced Bioengineering Initiative Center, K. N. Toosi University of Technology, Tehran 1417614411, Iran; 8Computational Medicine Center, K. N. Toosi University of Technology, Tehran 1417614411, Iran; 9Data Science and Artificial Intelligence Program, College of Information Sciences and Technology (IST), Penn State University, State College, Pennsylvania, PA 16801, USA; 10Department of Chemical Engineering, Faculty of Engineering, University of Waterloo, 200 University Ave W, Waterloo, ON N2L 3G1, Canada; 11Electrical and Computer Engineering Department, Sultan Qaboos University, Al-Khoud, Muscat 123, Oman

**Keywords:** nanoparticle, biodegradable, cornea, drug delivery, wound healing

## Abstract

During recent decades, researchers all around the world have focused on the characteristic pros and cons of the different drug delivery systems for cornea tissue change for sense organs. The delivery of various drugs for cornea tissue is one of the most attractive and challenging activities for researchers in biomaterials, pharmacology, and ophthalmology. This method is so important for cornea wound healing because of the controllable release rate and enhancement in drug bioavailability. It should be noted that the delivery of various kinds of drugs into the different parts of the eye, especially the cornea, is so difficult because of the unique anatomy and various barriers in the eye. Nanoparticles are investigated to improve drug delivery systems for corneal disease. Biodegradable nanocarriers for repeated corneal drug delivery is one of the most attractive and challenging methods for corneal drug delivery because they have shown acceptable ability for this purpose. On the other hand, by using these kinds of nanoparticles, a drug could reside in various part of the cornea for longer. In this review, we summarized all approaches for corneal drug delivery with emphasis on the biodegradable nanoparticles, such as liposomes, dendrimers, polymeric nanoparticles, niosomes, microemulsions, nanosuspensions, and hydrogels. Moreover, we discuss the anatomy of the cornea at first and gene therapy at the end.

## 1. Introduction

In 2020, the World Health Organization (WHO) announced that, 2.2 billion people in the world feel pain from vision blindness or impairment. Although various causes of vision impairment exist, the most important reasons are uncorrected refractive errors, cataract, age-related macular degeneration, glaucoma, diabetic retinopathy, cornea opacity, and trachoma. It is better to say that the main health problems in the elderly population are vision loss and blindness and it is useful to say more than 80% of people in the whole world are aged 50 years or older. Additionally, ocular disease treatment attracts particular interest due to the many chronic situations like age-related macular degeneration (AMD) and glaucoma, because it is clear that the eye is one of the most important issues in the body. The aggressive treatments include frozen therapy, laser therapy, etc., and these methods do not have acceptable effects on some patients with high recurrence rates [1].

The normal cornea is a dome-shaped, avascular, transparent connective tissue, and unique in a structure that covers the front portion of the eye and this precise tissue is one of the most densely innervated tissues in the body because it exposed to the external environment. This important tissue involving five layers (epithelium, Bowman’s membrane, stroma, Descemet’s membrane, and endothelium), and all of them refract light through the lens on the retina. Because of the different anatomical and physiological berries of the cornea, the efficient of the drugs is limited and it is the most important challenge for cornea drug delivery. There are different limitations, like tears, conjunctiva, sclera, blood–ocular barriers, and cornea structure, that exist for ocular drugs delivery. To overcome this problem, different drug delivery systems have been improved to increase the effectiveness of drugs for ocular treatment [1].

On the one hand, various methods like eye drops, ointments, contact lenses, punctual plugs, and have been used for corneal drug delivery. On the other hand, nanostructure materials are able to encapsulate and deliver small molecules so as to deliver them to various parts of the cornea. To improve the efficiency of the drugs and reduce the amount of precorneal drug loss, nanoparticles have to rapidly penetrate the cornea membrane and deliver especial drugs or small molecules to the specific part. Consequently, the drug could remain in prolonged contact with different parts of the cornea [2].

During the last decades, in overcoming the limitations of conventional methods, a significant amount of research has been undertaken to improve the efficiency of drug delivery, and each of them has some pros and cons. Different researches have been performed to engineer various kinds of nanoparticles, like liposomes, dendrimers, polymeric nanoparticles, niosomes, nanosuspensions, and hydrogels for corneal drug delivery. The main aim of using these nanoparticles is encapsulating, carrying, and delivering drugs, proteins, and peptides to specific parts of the cornea in order to enhance efficiency by improving drug penetration and reduce side effects because of the lower dosage requirements [3].

## 2. The Anatomy of the Cornea

The normal cornea is a dome-shaped, avascular, and transparent connective tissue that is unique in terms of its structure that covers the front portion of the eye, and this precise tissue is densely innervated because it exposed to the external environment [4]. This important tissue is supplied by sensory and autonomic sensitive nerve fiber. The horizontal of the cornea is 11.5 and vertical is 12 mm in an adult human. The thickness of this tissue is 0.5 and 0.7 mm at the center and in the sides, respectively. Additionally, this densely innervated tissue contains five different layers (epithelium, Bowman’s membrane, stroma, Descemet’s membrane, and endothelium) and all of them refract light through the lens on the retina, as shown in Figure 1 [5].

The epithelium layer consists of squamous cells and these kinds of cells create a refractive (70% of the total refractive power), transparent, and moist surface for the cornea [2]. The epithelium layer consists of 4–6 layers of corneal epithelial cells and it must be mentioned that the source of these cells is limbal epithelial cells, so the healing of minor wounds and self-healing properties are the two major properties of this layer because of these cells [5]. Various parameters have direct effect on the stem cells behavior and integration, the important one is substratum topography. Consequently, researcher have been trying to design biomimetic patterned for reconstructing of the epithelium cells. The next layer of the cornea after the epithelium is Bowman’s layer or Bowman’s membrane, and it has an acellular structure consisting of collagen fibers that do not conform to any definite pattern [6]. Moreover, nearly 85% of cornea thickness is due to the existing stroma layer, and as a result, this layer plays a critical role in maintaining the mechanical strength of the cornea. On the one hand, this connective tissue consists of 200–250 distinct layers of collagen fibers (named collagen fibrils). It should be mentioned that these fibers are arranged in parallel. Additionally, the stroma layer also be made of extracellular matrix (ECM), keratocytes, and glycosaminoglycans. The structure and the diameter of the collagen fibrils are two important parameters that have direct effect on the transparency of the cornea [7]. Next layer of the cornea structure called the Descemet’s membrane. Fibronectin, laminin, and collagen are three main components of this layer. This thick basement membrane is relatively tough because it composed of the meshwork of fibers and pores [8]. It should be noted that the Descemet’s membrane’s surface has a similar appearance to the cornea basement membrane but secreted by endothelial cells and resistance to proteolytic enzymes. Although this layer is tough, hearing stress can cause tear. At the end of cornea, an endothelium layer exists and it plays an important role in maintaining fluid transport, clarity of cornea, and vital tissue for stroma deturgescence because this layer pumps water into the stroma. This layer is composed of a single layer of endothelial cells and these kinds of cells have little capability to proliferate in adult cornea [9]. Finally, these cells are reduced during age or damage and the research focusing on the regeneration of endothelial cells is sparse because it’s difficult to maintain properties of the cells in vitro with proliferative capability [10].

## 3. Methods for Cornea Drug Delivery

Ocular disease treatment attracts particular interest due to the many chronic situations, such as age-related macular degeneration (AMD) and glaucoma, and the large markets for related therapeutics [11]. Moreover, because of the different anatomical and physiological berries, drug delivery for ocular tissue has continuously been challenging for drug delivery scientists and ophthalmologists. To design an ideal delivery scheme, drug should release controllable and the specific and target site and enhanced drug bioavailability should be considered [12]. Drug delivery could be carried out through dissimilar methods like topical, intravitreal, and transscleral depending on the origin of ocular disease. On the other hand, ocular drug delivery also categorized into anterior and posterior segment drug delivery. As it mentioned, cornea is the outermost tissue in the eye’s center and responsible for allowing the passing of light information to the retina so, its diseases can alter vision in many ways. The anatomical features of the cornea, the related barriers, and the drug delivery methods are shown in Figure 1. Among these barriers, there are the tear film and the cornea as physiological components. Physiological topical methods, drug loaded ocular inserts, and injection can be used for corneal drug delivery, and they are explained as below [11].

### 3.1. Topical Methods

The most patient compliant and convenient mode of drug delivery is topical application. The topical method is a perfect route of administration that can be proper for the handling of ophthalmic illnesses directly related to the anterior segment. Conversely, it becomes a challenging task due to some barriers such as the cornea epithelium, nasolacrimal drainage, metabolism in the eye, and blood-ocular barriers [13]. For anterior chamber diseases, in former times, ophthalmic drug delivery was used by topical formulations for ointments, emulsions, gels, solutions, or suspensions. Topical instillation presents many benefits, like slight absorption into the systemic circulation, noninvasiveness, prevention of first pass metabolism, relatively small dose, and ease of administration. On the other hand, major issues, such as contact through tear protein, rapid drainage, the tear turnover rate (1 μL min^−1^), quick metabolism, and rapid drug removal as a result of reflex blinking should be addressed for developing topical ocular drug delivery systems. Among topical formulations for gels, ointments and ophthalmic drug delivery eye drops are utilized for corneal drug delivery [14].

#### 3.1.1. Liquid/Solution Eye Drops

Drop instillation is one of the most important and common methods for careering different drugs into various parts of the eye, especially the precorneal segment. However, during eye blinking, and just 20% of the original applied dose is kept in this part of the eye. The concentration of the drug in the precorneal location determines the amount of drug diffusion. It must be noted that effective ocular drug distribution using eye drops is dependent on a greater retention time and elevated cornea permeation in the cornea tissue [15].

The most important properties of topical drops are safe, immediately active, convenient, patient compliant, suitable for the product, and, above all, noninvasive. As expected, after topical drug instillation, the concentration of the drug is high and after that concentration quickly decrease. For dissolving this problem, researchers have suggested to add different additives like viscosity enhancers, permeation enhancers, and cyclodextrins to topical eye drops so as to improve permeation, increase contact time of drug, and ocular bioavailability [16]. Through cornea integrity modification, permeation enhancers such as polyoxyethylene glycol ethers, sodium taurocholate, and etc. could enhance cornea uptake via cornea integrity modification [17,18,19]. The bioavailability of ocular drugs is improved by adding permeation enhancers to the ocular solutions. However, some local toxicity has been reported according to utilizing these enhancers [20]. Moreover, viscosity enhancers such as sodium carboxymethyl cellulose allow to reach some variation in essential variables like bioavailability and stability time in the precorneal areas with the aim of uptake improvement [21,22]. Hydrophobic drug molecules could be carried by cyclodextrines in an aqueous solution that facilitates the drug delivery to the biological membrane surface [23].

#### 3.1.2. Ointments

Ocular ointments as carrier systems, developed for the topical administration of drugs, consist of a compound of semisolid and a solid hydrocarbon (paraffin). The melting point of an ocular ointment is close to physiological ocular temperature (34 °C). To select the proper hydrocarbon, biocompatibility is the most important property that should be considered due to the prevention of more organic side reactions and warranting the usefulness of the formulation to sustain the process of delivery and to improve the bioavailability of the drug [15,16]. Ointments present some advantages compared to eye drops, like minimizing the tear dilution, more contact time, more effectual concentrations, and less nasolacrimal drainage. Blurred vision may be the result of using ocular ointments due to the difference in refractive index between tears and nonaqueous ointment base as an important disadvantage of ocular ointments. The base vehicle of the ointments should be comfortable for the eye and reconcilable with the packaging material and other ingredients. A combination of liquid petrolatum (mineral oil) and white petrolatum is broadly used as a vehicle. However, utilizing gels that dissolve in water, has recently increased since better stability, lubricity, pH, spreadability, and low irritability compared to petrolatum bases. Using some polymers, such as carbopol, carboxymethylcellulose, etc., for the preparation of gels could improve the contact time and consequently improve the ocular bioavailability because they have a mucoadhesive property [13].

#### 3.1.3. Ocular Gel

Another method to delivery various drug to different parts of corneal is using ocular gel. With this strategy, achievement of an optimal drug concentration at the target area is accessible [24]. Medical gels are an example of such systems that have attracted substantial attention [25]. Aqueous gels based on hydrophilic polymers (hydrogels) as well as those formulated using stimuli responsive polymers (gel forming systems or in situ gelling) have attracted growing interest for different applications related to eye health. Using these formulations, sustained levels of bioactives and drugs could be achieved at target sites by combination of different ophthalmic pharmaceuticals. Some polymeric gels that have in situ properties, as liquid dosage forms, have been recently developed to improve ocular drug delivery. These systems face phase transition on conjunctival cul-de-sac or the ocular surface to make a viscoelastic gel in response to environmental stimuli. These stimuli can be classified as chemical, physical, and biological, while each of them use a different mechanism to form the gels. These triggers have the ability to be altered to control residence time, gel formation, and ocular drug release. In situ polymeric gelling systems do not require copolymerisation agents or organic solvents that allow them to be more interesting. Semi-synthetic or natural polymers could be utilized in preformed ocular gels. Those utilized to produce in situ gelling systems could be pH-responsive, ion-responsive, or temperature-responsive (thermoresponsive) based on the environmental stimuli triggering the transformation of these polymers from sol to gel [26].

### 3.2. Drug Loaded Ocular Inserts

Ocular inserts, as solid devices placed in the conjunctival cul-de-sac, are designed with the aim of drug releasing at a permanent rate for an extended duration of time while improving patient compliance and lessoning systemic absorption to the minimum amount through the nasal mucosa. The first marketed device was the pilocarpine Ocusert^®^ (ALZA Corp., Palo Alto, CA, USA) that reach zero-order kinetics. In this device, a reservoir encompassed by a pair of release-controlling membranes was used to carry the drug. It was showed that the method had a therapeutic effect with less amount of drug and therefore lower side effects [27]. Despite the many advantages of ophthalmic inserts, they have some disadvantages, such as their solidity, as the major one, that causes the patients to feel it as an extraneous body in the eye, the movement around the eye, the interference with vision, the occasional inadvertent loss while rubbing the eyes or during sleep, and the difficult placement of them [28].

#### 3.2.1. Contact Lenses

Eye drops remain widespread to treat ocular diseases. However, because of reduced patient compliance and low bioavailability, there is a requirement to shift to drug-eluting non-invasive therapeutic contact lenses (CLs) [29]. Medicated contact lenses are one of the ocular inserts broadly utilized for therapeutic applications, aesthetics, and correcting vision. CLs are utilized as therapeutic devices for ocular surface and anterior chamber disorders. Moreover, it has been reported that CLs were applied as posterior segment disease management [30]. Anti-inflammatory, antimicrobials, immunosuppressants, corticoids, and ocular pressure lowering agents can be loaded in contact lenses and used for the treatment of various diseases [31]. These CLs should have cytocompatibility, comfort, and prolonged release. In fact, these drug carriers increase bioavailability, extend wear time, and improve retention time. Due to the solid formulation of contact lenses, the retention time can be extended to more than 30 min while this time is 1–3 min for drug delivery by eye drops. Hence, ocular bioavailability is also enhanced whereas dose frequency is removed and the drug absorption and delivery to the target ocular tissues occur. Moreover, the systemic absorption of the drug is avoided and also the related side effects are decreased [30]. Immersing the contact lenses in a solution of a particular drug is the usual method for loading that drug supply on them. According to the former studies, a superior profile of drug release is one of the advantages of immersed contact lenses over eye drops systems [32]. Furthermore, loading drugs on CLs have been performed using many technologies instead of immersing them in the drug solution. For example, according to a study, a novel drug-eluting contact lens that embedded lidocaine-laden nanoparticles was advanced and the observation of sustained in-vitro lidocaine release during 7–8 days was performed [33]. Likewise, another group produced a drug-eluting contact lens utilizing a polymer embedded matrix for econazole and ciprofloxacin. A drug release profile with zero-order was demonstrated by in-vitro data, which cause drug release to sustain up to one month [34,35]. Figure 2 shows the superiority of soft drug loaded contact lenses in comparison to eye drops. A successful illustration of extensively higher efficiency and safety over conventional eye drops is the major challenge about this therapy that should be addressed. Microbial resistance, oxygen diffusion, and continuous and effective drug release are also the issues that should be solved for successful commercialization for CLs [12].

#### 3.2.2. Punctal Plugs

Punctal plugs (PPs) are inserted in the tear ducts, as biocompatible devices, and occlude tear drainage. Their size is 2–5 mm and are known as acrimal plugs or occludes. Punctal plugs can release drugs to the anterior segment of the eye through a controlled and non-invasive manner [11,12]. After punctum blocking, the accumulation of tear fluid increases and the eye remains moist. By using PPs, tears are blocked through the canaliculi that is a link between the nose and the eye (Figure 3) [36]. These ocular inserts can be provided from non-biodegradable and biodegradable materials. Non-biodegradable punctal plug delivery system (PPDS) can be made from polycaprolactum, hydroxyethyl methacrylate, and silicone that can present controlled drug release till 180 days [37]. PPDS proposes many advantages over topical drug delivery like patient compliance, decreasing the lacrimal drainage of drugs, capability of attaining controlled drug delivery, and possibly lower costs [38].

### 3.3. Injection

According to the issues of conventional drug delivery, like the need for repeated drug administration, it should be necessary to administer novel drug delivery systems for sustained and controlled drug delivery. Among novel drug delivery systems, injectable formulations are highly impactful. Conventional hypodermic needles are utilized to deliver the drug via intraocular injections. Different routes are available for performing the intraocular injections, such as intravitreal, subconjuctivial, intracornea, periocular, intrastromal, intracameral, etc. [39]. Among these, intrastromal, subconjunctival, intracornea, and intracameral methods are exploited to manage the diseases related to the anterior segment of the eye. Using these injections, great drug concentrations could be achieved in the particular tissue. Subconjunctival injection is the most patient friendly one and directly administer the medication in the target sit (Figure 4). Hypodermic needles with the size of 21–30 G are commonly utilized for subconjunctival injections. This type of injection represented several side effects like hemorrhage in spite of its better performance compared to topical eye drops. Similarly, intracameral, intrastromal, and intracornea injections direct the medication to the target tissue. For instance, for effective drug delivery particularly for fungal keratitis and CNZ treatment, intrastromal injections are extremely useful. Despite promising results, intrastromal injection with a hypodermic needle is unpleasant for patients, highly- invasive, and painful. Also, they may cause bacterial infections, tissue damage, and inflammation [40]. In general, the intraocular injection has some disadvantages, including invasive nature, non-compliance, and less bioavailability due to the frequent application of them [41].

Microneedle (MN) is an attractive technology for drug delivery to the eye in a minimally invasive manner. MNs are devices with dimensions of few micrometers to 200 μm and made up of metal or polymer. Due to their micro-sized projections, they are minimally invasive in nature. The MNs can overcome the issues of conventional delivery systems and also cross the ophthalmic barriers to transport the drugs to the target site [42]. These micron-sized needles can easily insert on the eye for different applications. In comparison to the conventional hypodermic needles, MNs provide more patient compliance and can release the particular drug during a time period. Among the different types of MNs, hollow, dissolving, and solid coated MNs have a significant role in ocular drug delivery [43]. Literature indicates that hollow or single solid MNs have been used for ophthalmic drug delivery by drug molecules with different molecular weights including sustained-release depot forming gels, nanoparticles, or microparticles [44]. For example, according to a study, a detachable biodegradable MN was developed for minimally and long-term corneal drug delivery (Figure 5). It was a hybrid MN that consisted of a supporting base and a drug-loaded biodegradable tip. Impact insertion was required to apply the MN to the cornea. After application, it left the tip in the cornea so the drug could be released for a certain period. To attain an effectual intracorneal injection, insertion depth, the drug-tip dimension, and injection dwell time was precisely optimized. An Acanthamoeba keratitis model was used to apply the detachable hybrid MN and the process of delivering the drug tip to the cornea of a mouse was successfully performed in vivo. The therapeutic efficiency of this method was confirmed through the follow-up of the managed cases for a week [45].

## 4. Nanostructured Materials and Properties

The aims of delivering the drugs to the anterior segment of the eye based on the therapeutic target areas can be classified into two categories: (a) to enhance drug bioavailability at extraocular tissues to diminish the symptoms and signs caused by ocular surface inflammatory disorders of cornea and conjunctiva, like allergic diseases and eye syndrome, and (b) to improve drug bioavailability in the intraocular tissues for treating vision-threatening, infections and complex diseases, such as intraocular inflammation (uveitis) or glaucoma [46,47]. Various factors limiting bioavailability of ocular drugs for patient such as: tears, conjunctiva, sclera, blood-ocular barriers, and cornea structure. According to the recent advances in nanotechnology, novel opportunities have been provided to face the challenges of conventional drug delivery systems. In this regard, nanostructures that are able to encapsulate and deliver small molecules have been fabricated [48]. During past two decades, nanoparticles are being investigated with the purpose of improving drug delivery systems for ocular. In scientific societies, materials with size between 1 and 1000 nm in at least one dimension called nanoparticles and it must be noted that using larger size particles (more than 1000 nm) for ophthalmic drug delivery have negative effect on treatment and caused ocular irritation [45,49]. Ocular drug delivery systems are so exciting and challenging because of anatomy of this tissue, critical and specific environment that exist in this important tissue. For having better drug delivery systems, nanoparticles have to rapidly penetrate the cornea membrane so as to deliver especial drugs and as result, drug could remain in prolonged contact with different parts of the ocular. Furthermore, the best drug delivery system has to reduce amount of precorneal drug loss [50]. Recently, researcher have been tried to fabricate various kinds of nanoparticles for cornea drug delivery with the purpose of encapsulating, carrying, and delivering drugs and small molecular to specific parts of the cornea. Additionally, cornea drug delivery has been used for delivering various kinds of drugs, proteins, and peptide to cornea so as to control the release rate of them and specific location in order to enhance efficiently by improving drug penetration and reduce side effects because of needing lower dosage requirements [51,52].

Different parameters have direct effect on the properties of nanocarriers and they must be considering before designing and using for cornea drug delivery. As an illustration, surface charge of nanoparticles has to be cationic for more efficiently because the surface of the cornea is negative. As a result, nanocarriers with cation surface charge are able to increase the retention time and creating an enhanced opportunity for the therapeutic agents to come to the eye [53]. Another key factor for designing of nanocarrier for cornea drug delivery is size, because this parameter has direct effects on the amount of penetration into cornea tissue [54]. Additionally, several studies have shown the best size of nanocarrier for cornea drug delivery, and nearly all of them have mentioned size of nanocarrier have to be less than 200 nm in order to simply taken up in the conjunctiva and cornea because of existing ocular barriers [15].

By using nanoparticles as nanocarrier, it is possible to deliver drugs, small molecular or entities to the posterior and anterior segment ocular [55]. Various kinds of nanoparticles have been introduced by researchers for cornea drug delivery and all of them have important and unique pros and cons [56,57]. Additionally, different key approaches are suggested to improve ocular bioavailability. However, the best way for increasing efficiently and improving drug penetration is using biodegradable nanocarrier for cornea drug delivery because they have shown satisfactory ability for cornea drug delivery system and by using this kinds of nanoparticles, residence time of the drug enhanced in various part of the cornea, drug can release sustainably, nanocarrier degrade after specific time and, etc. [58]. A variety of nanocarriers, such as liposomes, dendrimers, polymeric nanoparticles, niosomes, nanosuspensions, and hydrogels, have been advanced for ocular drug delivery. In Section 5, such nanocarriers will be described thoroughly.

## 5. Nanoparticles as Biodegradable Nanocarriers for Cornea Drug Delivery

### 5.1. Liposomes

Liposomes are spherical lipid vesicles consisting of at least one phospholipid bilayer which encloses an aqueous core [16]. They can be made in various sizes from 0.08 to 10.00 µm. The physiological barrier function of the cornea poses a challenge and hinders deeper ocular drug permeation. Thus, liposomes have proven able to circumvent this limitation and provide the possibility of drug delivery to both anterior and posterior segments of the eye in a sustained manner [59]. Excellent biocompatibility, the capability of loading both hydrophilic and hydrophobic drugs, and cell membrane like structure make liposomes ideal candidate for ocular drug delivery purposes [60]. Taking all of this information into account, recent progress in liposomal cornea drug delivery has been summarized in Table 1.

A plethora of studies have been published with the main aim of demonstrating the effectiveness of liposomes for cornea drug delivery applications. Law et al. investigated the in vivo and in vitro cornea penetration and absorption of acyclovir delivery from acyclovir-loaded liposomes for herpes keratitis treatment. Forming a cover layer on the surface of the cornea, positively charged liposomes provided slow penetration of acyclovir through cornea and increase of its absorption [61]. The objective of the study done by Dai et al. was evaluated of the performance of Tacrolimus-loaded liposomes containing bile salts for increasing cornea permeability. Results of ex vivo cornea transport and in vivo cornea uptake indicated that liposomes containing bile salts can increase the cornea permeability by 3–4 fold comparing with conventional liposomes containing cholesterol [62,63]. In another study, Li et al., interpreted liposomes as a novel approach for brinzolamide delivery. The measured cornea permeability demonstrated that brinzolamide-liposomes had a high transmembrane permeation ability which resulted in improving the brinzolamide transport through cornea. Furthermore, an effective and sustained interocular pressure reduction can be obtained by liposomal brinzolamide delivery [64]. Silk fibroin-coated ibuprofen-loaded liposomes were fabricated by Dong et al. Silk fibroin formulation was selected for coating liposomes for improvement of cell adhesion and uptake behaviors. Sustained drug release and high transcorneal permeability along with no cytotoxicity were achieved by these silk fibroin-coated liposomes [65]. Since peptide delivery is really challenging, due to their instability and enzymatic degradation, several studies have tried to use liposomes for peptide delivery through cornea. For example, Soriano-Romaní et al. prepared thrombospondin-1-derived peptide, KRFK-loaded liposomes for transcorneal permeability improvement [66], and in another study, Neves et al. proposed anionic liposomes for peptide and cDNA delivery to cornea epithelial cells. Anionic liposomes with the size of 138.8 ± 34 nm and charge of −18.2 ± 1.3 mV were prepared. Peptides were efficiently delivered to human cornea epithelial cells through liposome-based peptide delivery. Moreover, the use of anionic which was complexed with plasmid DNA culminated in a modest while advantageous transfection of human cornea epithelial cells [67]. The preparation procedure of these liposomes is schematically shown in Figure 6. To reduce the limitations associated with the use of solid liposomes, deformable chitosan-coated flurbiprofen-loaded liposomes were proposed by Chen et al. [68]. These flexible liposomes offer some advantages. First, they are able to easily move across the pores which are too small for conventional ones to pass, as their bilayers represent a high curvature. In addition, precorneal retention and ocular biocompatibility improvement were achieved by these flexible liposomes [68]. Liposomes have also proven to be a good option for delivery of anti-inflammatory agents to cornea epithelial cells for inflammation-based eye diseases. In a study performed by Soriano-Romaní et al., medroxyprogesterone acetate (MPA)-loaded liposomes were prepared. MPA-loaded liposomes attaching to the cornea epithelial cells to provide continuous sustained drug release without any side effects [69].

It should be mentioned here that, in some eye diseases, like fungal infections, prolonged drug delivery is needed, so liposomal drug release would be the best answer [70]. To this aim, Moustafa et al. designed hyalugel-integrated liposomes for enhancing corneal permeability and long-term fluconazole delivery. In continuation of their previous study, this time they prepared and evaluated fluconazole-loaded carbopol-integrated gel-core liposomes (carbosomes) for ocular drug delivery purposes [71].

An ion pair technique was used by Ren et al. for the preparation of azithromycin-cholesteryl hemisuccinate ion pair liposomes (ACIP-Lip). The entrapment efficiency and drug loading capacity were improved due to the use of this method. Their results showed that continuous and pH-sensitive drug release was obtained through the use of ACIP-Lip [72]. Others researches are summarized in Table 1.

**Table 1 pharmaceutics-12-01232-t001:** Advancement in liposomal for cornea drug delivery.

Kinds of Drug	Result
Acyclovir loaded liposome for herpes keratitis treatment	positively charged liposomes provided slow penetration of acyclovir through cornea and increase of its absorption [73]
Tacrolimus-loaded liposomes containing bile salts	liposomes containing bile salts can increase the cornea permeability by 3–4 fold comparing with conventional liposomes containing cholesterol [62]
Brinzolamide-loaded liposomes	high transmembrane permeation ability, improvement of the brinzolamide transport through cornea, effective and sustained interocular pressure reduction [64]
Silk fibroin-coated ibuprofen-loaded liposomes	improvement of cell adhesion and uptake behaviors, Sustained drug release and high transcornea permeability, no cytotoxicity [65]
thrombospondin-1-derived peptide, KRFK-loaded liposomes	transcornea permeability improvement [66]
anionic liposomes for peptide and cDNA delivery	efficient peptide delivery to human cornea epithelial cells, modest while advantageous transfection of human cornea epithelial cells [67]
deformable chitosan-coated flurbiprofen-loaded liposomes	easily moving across too small pores, precornea retention and ocular biocompatibility improvement [68]
medroxyprogesterone acetate (MPA)-loaded liposomes	continuous sustained drug release without any side effects [69]
fluconazole-loaded carbopol-integrated gel-core liposomes (carbosomes)	enhancing cornea permeability and long-term fluconazole delivery [74]
azithromycin-cholesteryl hemisuccinate ion pair liposomes (ACIP-Lip)	improvement of entrapment efficiency and drug loading capacity by the use of ion pair method, continuous and pH-sensitive drug release [72]
tetrodotoxin and dexmedetomidine-loaded modified succinyl-Concanavalin A liposomes	long period of analgesia, sustainable release of both tetrodotoxin and dexmedetomidine, modified liposomes were persisted on the surface of the cornea for long time duration [75]
chitosan coated liposomes Triamcinolone Acetonide delivery	sustainable and reliable Triamcinolone Acetonide delivery, high drug encapsulation efficiency along with high positive surface charge [76]
polyamidoamine dendrimer (PAMAM G3.0)-coated compound liposomes for berberine hydrochloride	enhancement of the bioavailability of berberine hydrochloride with no side effects [77]

In the case of local ocular analgesia, a prolonged, minimally toxic drug administration process is clinically desirable, in particular for ophthalmic surgery, whereby frequent drug administration could have numerous negative consequences. Zhan et al. showed that a long period of analgesia, 105 min complete, and 608 min partial analgesia and sustainable release of both tetrodotoxin and dexmedetomidine are achievable through the use of modified succinyl-Concanavalin A liposomes. In fact, these modified liposomes were persisted on the surface of the cornea for the long duration [75]. As mentioned earlier, liposomes exhibit great potential for drug delivery through cornea to the posterior segment of the eye. In a study conducted by Khalil et al., they tried to overcome the limitations related to the posterior eye drug delivery, by proposing chitosan coated liposomes for more sustainable and reliable triamcinolone acetonide delivery compared to the conventional liposomes, see Figure 7. Moreover, this type of chitosan coated liposome demonstrated nice drug encapsulation efficiency along with high positive surface charge. Note that the cornea surface has a negative charge, and this feature would be helpful in terms of cornea drug delivery applications [76]. One of the main reasons behind the application of the liposomes for ocular drug delivery is the instability and poor bioavailability of that specific drug compound. Lai et al. prepared polyamidoamine dendrimer (PAMAM G3.0)-coated compound liposomes to enhance the bioavailability of berberine hydrochloride with no side effects [77].

### 5.2. Dendrimers

Dendrimers are tree-like nanosized polymeric drug delivery systems, Figure 8. There branched nanosized systems can be found in different molecular weight with terminal end consisting of different functional groups of amin, carboxyl, and hydroxyl. A highly branched shape with different functional groups, nanoscale structure, the possibility of being made in different molecular weight, and surface charge, make these carriers to be as a promising tool for target delivery of both hydrophilic and hydrophobic drugs [16,78].

In a study done by Souza et al. firstly the effect of iontophoresis on polyamidoamine dendrimer permeation and distribution in cornea tissue was investigated [79]. Secondly, they also tried to assess the impact of iontophoresis on the amount of dexamethasone that permeated by dexamethasone-PAMAM complexes across cornea. In this study, they investigated the distribution of fluorescein isothiocyanate labeled anionic (G3.5) and cationic (G4) dendrimers for different application times. Results suggested that the distribution of dendrimers in cornea depends on the iontophoresis application time. Improving the aqueous solubility of dexamethasone which was achieved by complexing the dendrimers with dexamethasone and increase of its concentration in aqueous humor obtained by iontophoresis were other results of this research. It was demonstrated that application of iontophoresis in association with dendrimers could greatly improve the intensity and permeability of dendrimers across cornea and also good interaction between drug and dendrimers guaranteed the bioavailability of the drug [80]. Dendrimers also pave the way for drug delivery to the posterior segment of the eye like the retina, according to what was discussed by Yavuz et al. PAMAM dendrimers complexing with dexamethasone effectively improved the dexamethasone permeation through both cornea and sclera [81].

It is worthwhile to note that, dendrimers offer a high flexibility for sustainable and targeted cornea drug delivery through enhancement of its permeability for different model drugs and clinical applications, such as development of mildly cross-linked dendrimer hydrogel for antiglaucoma delivery [82], or the incorporation of an injectable dendrimer-dexamethasone gel formulation for targeted cornea inflammatory treatment [83].

### 5.3. Niosomes

Niosomes (microscopic lamellar structure) are biodegradable, nonimmunogenic drug carriers with a bilayer structure that are constituted by nonionic surfactants with cholesterol in an aqueous media [84]. They are potentially utilized for drug delivery in a targeted and controlled manner. Niosomes variation can be determined by their method of production and structure of their bilayer. Generally, they can be formed in different types of small unilamellar vesicles, multi lamellar vesicles, or large unilamellar vesicles. There are several advantages which make these carriers a good candidate for ocular drug delivery applications [85].
-The capacity for the entrapment of both hydrophilic and lipophilic drugs.-Providing drug administration in a sustained and controlled manner, as vesicles work as a storehouse.-Increasing the bioavailability of drugs.-Having a better stability compared to liposomes.-Osmotically active.-Nonimmunogenic to body.

Several studies have tried to demonstrate niosomes capabilities in terms of cornea drug delivery applications. Kaur et al. conducted research with the aim of establishing a niosomal based timolol maleate (TMREVbio) delivery. They showed that use of niosomes for timolol maleate delivery effectively improves in vitro cornea permeability plus interocular pressure (IOP) reduction effect, with sufficiently high concentration of the drug in aqueous humor [82]. Abdelkader et al. prepared a niosomal formulation for naltrexone hydrochloride (NTX) delivery through the reverse-phase evaporation (REV) method. These niosomes possess the ability to hold NTX up to 61.5% (*w*/*w*), and another point is that the diameter and the general shape of these carriers were dependent on the additives used for their membranes meaning that, their size and the morphological shape of them was flexible. For example, adding F-C24 as the additive resulted in a giant oval shape of niosomes, called discosomes, while the typical shape of these niosomes was spherical. These giant niosomal drug carriers offered some benefits, first of all, based on their big size the risk of drainage through nasolacrimal duct would be diminished, besides, their oval shape improved the precornea retention. Controlled NTX delivery with the enhancement of cornea permeability was of the significant advantages of these prepared niosomes [86]. Four niosomal drug carriers were assessed in another study by Abdelkader et al. for corneal toxicity by the hen’s egg test chorioallantoic membrane (HET-CAM), bovine cornea opacity and permeability (BCOP), and cornea histological test. They claimed that these testes could be as an alternative for the Draize test [87]. In another study, Elmeshad et al. investigated the capability of spanlastics niosomes including two edge activators (Tween 20 or 80) for itraconazole delivery. Owing to the flexibility of these vesicles they can be used for the safe drug delivery to the anterior and posterior parts of the eye. The application of this formulation of the niosome was accompanied by the enhancement of the transcornea permeation [88].

Two studies developed two different niosomal based drug delivery approaches for tacrolimus (FK506) delivery. In the first of these two studies, Li et al. prepared proniosome-derived niosomes. The reformed niosomes were prepared by the tacrolimus loaded proniosome, and the process is precisely elaborated in the original paper. Based on their results, the application of FK506 loaded niosomes was resulted in the postponement of the cornea allograft rejection process, in addition to other advantages like improvement of cornea permeation and retention of FK506 [89]. In the other study, hyaluronic acid (HA) coated niosomes were prepared with the aim of making improvements in aqueous humor pharmacokinetics, precornea retention, and cornea permeability. Moreover, it was shown that HA-coated niosomes demonstrated stronger adhesiveness to mucin compared to the noncoated one [90].

Besides all the advantages attributed to the niosomal based ocular drug delivery applications, improvement of cornea permeability and drug availability, they seem to not be efficient enough for entrapment of hydrophilic drugs. Regarding this fact, Dehaghi et al. proposed two loading methods, passive and remote. Some factors were considered as variables in each trial. Results reveal that the encapsulation efficiency of dorzolamide loaded in niosomes increased by increasing both the lipid concentration and cholesterol percentage. Based on this study, the remote method was more efficient than the other one [91]. In a research study conducted by el-Nabarawi et al., natamycin (NAT) loaded niosomes in ketorolac tromethamine (KT) gels were designed in order to evaluate their performance through improvement of pharmaceutical effects which would be mirrored in the enhancement of cornea permeability along with the reduction of the inflammation effect of fungal keratitis. It was concluded that F8 (NAT loaded niosomes/0.5% KT-4%Na.CMC) potentially offered the best performance and provided a sustained natamycin delivery [92].

### 5.4. Nanosuspensions

Nanosuspensions are colloidal dispersions of nanoparticulate drugs with stabilizers, which enhance the solubility of lipophilic drugs owing to the nanoscale structure and large surface area. Nevertheless, limited researches had been done and more attention needed in this area [93]. Entrapping nanosuspensions into hydrogels as contact lenses would help to enhance drug loading capacity and more sustained release. pHEMA as soft contact lenses with high retention of water in their network and poorly water-soluble triamcinolone acetonide nanosuspensions can enhance residence time of drug on the cornea surface and more controllable drug release, due to good adhesion to cornea and acting as a drug reservoir. Puloronics and PVA as stabilizers could give steric stabilization and form a hydrophilic layer at the surface of the nanoparticle, respectively. Release profiles showed that pHEMA lenses have more sustained drug release in 48 h, in comparison to SCL (Commercial soft contact lenses) release in 12 h [94]. In another study for an allergic eye, OLO (olopatadine hydrochloride- antihistaminic effect on mast cells)-loaded Kollidon^®^ SR nanoparticles with spray drying as a non-toxic and cost-effective method have been used. Drug loading efficiency and encapsulation efficiency depended on polymeric and drug concentration to be loaded (12.21% and 60.78% at best, respectively). The release profile showed maximum release of 90% in 48 h, while without nanosuspension, time reached 3 h. In addition, the retention study revealed that nanoparticles were washed off the cornea surface within 24 h, but the retention time of drugs has been increased rather than a solution without nanoparticles. Prolonged drug release and retention enhanced patient compliance [95].

A novel block copolymer nanosuspension that consist of chitosan (CS) and methoxy poly (ethylene glycol)-poly(ε-caprolactone) (MPEG-PCL) has been developed which could accept hydrophobic diclofenac (DIC) through self-assembling into cationic micelles. In vitro release study showed that sustained release of 73% occurred at 8 h. This release influenced by such factors like the amount of drug diffusion and efficiency, polymer degradation, and interaction between carrier and drug. Drug release profile and drug pre-cornea penetration have been shown in. It can be seen that burst release even occurred in DIC/MPEG-PCL-CS, but from clinical aspects, within this initial release, the concentration of the drug, quickly reach to a necessary amount, before sustained release. Cornea penetration in DIC/MPEG-PCL-CS was 1.4 times greater than commercial eye drop. Furthermore, in vivo penetration test showed higher fluorescence intensity in Nile red MPEG-PCL-CS nanosuspension in comparison to Nile red aqueous solution, suggested the enhancement of penetration. Although an ocular irritation test revealed temporary irritation according to cationic nature of the polymer, which recovered after 6 h, no further irritation was observed after the 24-h instillation. The higher concentration of DIC (Cmax) in the aqueous humor in DIC/MPEG-PCL-CS observed from the area under the aqueous humor concentration curve rather than commercial eye drop (about 2.3 times larger) [96].

### 5.5. Hydrogels

Hydrogels are hydrophilic 3D networks that can absorb more than 100% of their dry weight. Hydrogels can be divided into natural, synthetic, or hybrid compositions [97]. Crosslinking is about linking one polymer chain to another via a bond, through a physical or chemical process. The multidimensional network structure and stability of hydrogels are the result of these bonds. Physical properties of hydrogels such as solubility, elasticity, and viscosity can be affected by the process of cross-linking [98]. Permeability and diffusion properties of these networks can be tuned by differing processing parameters like temperature, type of crosslinker, crosslinking time, the ratio of polymer and crosslinker. In comparison to hydrophobic polymers such as PLGA or PLA, hydrogels are typically formed at ambient temperatures without requiring harsh processing conditions or organic solvents, which may denature antibodies or proteins. An important feature in ocular drug delivery is their ability to absorb and retain water or other aqueous solutions. So, hydrogels are promising candidates for the entrapment of fragile entities, including antibodies, proteins, cells, peptides, and oligonucleotides, due to the mentioned forming process and existence of an aqueous internal environment. Hence, one of the appropriate applications of hydrogels is in contact lenses [94].

In ocular applications, hydrogels have been explored for sustained drug delivery, periocularly as well as intravitreally [99]. Hydrogels also play an essential role in nanotechnology developments for ocular drug delivery. The formulation of hydrogel nanoparticles is an example of the applications of hydrogels in nanotechnology. These systems benefit from both hydrogel properties and the little size of a nanoparticle [98]. Viscous nature of hydrogels presents longer retention of entrapped medicaments at the targeted site and their transparency diminishes blurring of vision. For these reasons, they have been broadly investigated for the applications of drug delivery to the eye. Moreover, the gel structure and consistency of hydrogels allow them to entrap nanoparticulate delivery systems. These systems can potentially enhance bioavailability in comparison to hydrogels without nanoparticles. Therefore, hydrogels can be utilized for intraocular or topical drug administration so as to decrease drug clearance by dynamic ocular barriers [100]. For example, for the topical administration of nanoparticles, they should be incorporated into a suspension system. For the base of this system, the hydrogel is proper. Therefore, while nanoparticles help for permeation of the drug through the eye barriers, the polymeric hydrogel cause mucoadhesion increasing to improve bioavailability. In this regard, such a system has been developed, with the presence of levofloxacin, that indicated extended cornea residence time along with increased antibacterial activity in comparison to using a levofloxacin solution [101,102]. As another example of using nanogels for ocular drug delivery, dexamethasone loaded pH-sensitive, ion-sensitive, and temperature-sensitive nanogels were formulated to improve cornea permeability and the ocular surface retention of the encapsulated drug. Despite high capacity to load drugs, cornea permeability and the ocular surface retention of traditional nano eye drops are yet disappointing and thus they restrict therapeutic efficacy. These prepared nanogels presented good viscosity and strength under physiological conditions as well as excellent rheological properties. Among these nanogels, temperature-sensitive gel represented the best behavior. The stimuli-responsive nanogels can be considered as stable systems that are useful for sustained and targeted drug delivery to the posterior segment of the eye [103].

Nanogels also play a major role in drug delivery to the cornea. For instance, for the treatment of corneal infections, tenoxicam nanogel was developed and optimized. The optimization was performed using the Box-Behnken experimental design. The optimized formulation of tenoxicam nanogel presented a particle size of 203.18 nm and indicated 92.30% entrapment efficiency. According to the in-vitro release study, extended drug release was achieved with the optimized formulation. The results showed that tenoxicam nanogel can be an efficient carrier for delivering tenoxicam and other drugs to the eye [48]. According to another study, Swilem, et al. used a strategy related to nanotechnology to provide a low-viscosity poly (acrylic acid) (PAAc)-based to enhance patient compliance and efficiency. Chemically crosslinked PVP/PAAc nanogels were produced by self-assembly between PAAc and PVP via hydrogenbonding along with gamma ray induced cross-linking. By controlling the irradiation dose and feed composition, the optimization of mucoadhesiveness, transparency, and viscosity of nanogels was achieved. Moreover, using a dry eye model, it was demonstrated that indications of eye dryness were improved more rapidly and efficiently (three days) using PAAc-rich nanogels produced at 20 kGy in comparison to highly viscous commercial Vidisic^®^ gel (Figure 8). The progressive absorption of these nanogels at the cornea surface could be facilitated through mucoadhesiveness and nanoscale size of them. These nanogels can lessen the symptoms of dry eye efficiently and with more patient compliance. Also, the new generation of artificial tears can be introduced by these nanogels [49].

At the ocular surface, antioxidant levels are important for wound healing of cornea after a chemical injury or trauma. However, due to the low stability, low residence time, and poor solubility, the bioavailability of efficient antioxidants like ferulic acid is restricted after topical delivery. In a recent study, ferolic acid has been formulated in a nanocomposite platform consisted of micelles and nanogels for effictive drug delivery to the cornea. Hyaluronan was cross-linked with ε-polylysine after adding to the ferulic acid loaded and blank micelles. Hyaluronan nanogels showed positive zeta potential values with dimensions of ~300 nm. Characterization of the formulations was performed in terms of wound healing properties, biocompatibility, rheological behavior, release pattern of ferulic acid, and its penetration into excised bovine corneas. The drug release with micelle-nanogel composites is sustainable up to two days compared to Pluronic^®^ micelles that rapidly released ferulic acid. Moreover, after micelle-nanogel application, ex vivo penetration investigations indicated an appropriate accumulation of ferulic acid into excised bovine corneas. These systems have shown promising results for the treatment of corneal wounds due to the wound closure properties and the good cytocompatibility of the micelle-nanogel formulations [50].

For the topical administration of ophthalmic drugs, intraocular pressure has been always challenging. In this regard, for effictive topical delivery of acetazolamide (ACZ), ocular surfactant based nanovesicles (NVs) that transfered in mucoadhesive nanogel was developed through a study in 2019. A variation of Chitosan-sodium tripolyphosphate (CS-TPP) nanogel concentrations was used to embed the ACZ loaded nanovesicles (ACZ-NVs). The most promising viscosity, rheological behavior, and adhesion time were offered by the nanogel prepared by utilizing 1.5% CS. According to the in vitro release of ACZ, the obtained results represented a controlled release profile after using nanogels. Also, minimal irritation was reported for this formulation in comparison to ACZ topical suspension according to the in vivo irritation test. Utilizing an ACZ-NV nanogel formulation, IOP decreased gradually, achieved the normal value after 45 min, and then represented extended action to 24 h. This is the most inspiring result among these three formulations of acetazolamide shown in Figure 8 and also ACZ oral tablets (Commercial ACZ). The reason was the mucoadhesive action of the nanogels that for a long time kept the drug in contact with the ocular tissue. This research work indicated that mucoadhesive nanogel for ACZ drug delivery can be promising for further applications [51].

Nanostructured lipid carriers (NLC) that, owing to their good penetration, control drug release, and retention of drugs in a solid matrix that avoids the agglomeration of nanoparticles, have attracted considerable attention as carriers in ocular drug delivery. However, because of their low viscosity, NLCs have rapidly expulsion, which limits their applications. Novel hybrid hydrogen based on carboxymethyl chitosan (CMCS) and poloxamer 407 (F127) has been fabricated, using genipin as a non-toxic cross-linker. pH and temperature of the environment had an effect on the swelling ratio of the hydrogel. Represents the in vitro release study of baicalin (BN), an ophthalmic anti-inflammatory drug, in a BN-drop eye, BN-NLC, and BN-NLC gel with a CMCS-to-F127 weight ratio of 3.0:1.0. Test results revealed 81.10% after 10 h in BN-NLC gel, showing more stable release rather than BN-drop eye, BN-NLC. Proved the most cornea penetration for BN-NLC gel with the highest apparent permeability coefficient among the others [104]. For cannabigerol acid (CBGA) with limited cornea penetration (<5%), embedded nanoparticles (poly (ethylene oxide) (PEO) and poly (lactic acid) (PLA) as drug nanocarriers) in hydrogel as a composite of hyaluronic acid (HA) and methylcellulose (MC) has been formulated. Kabiri, et al. designed an experiment (33-factorial design that assessed relations of three independent variables (i.e., concentration of HA and MC and size of nanoparticles) with the transition as dependent variable to determine sol-to-gel transition point (temperature) in a formulation which can switch its rheology between thixotropy and temperature-dependent rheopexy. At this point, the hydrogel can rapidly turn to gel near the temperature of the cornea surface. The rheology property of hydrogel was very close to the temperature of the cornea surface. Moreover, the amphiphilicity of nanoparticles, their size, and spherical morphologies, altogether, enhanced the stability, penetration, and reduced irritation of drugs. SEM and AFM analysis proved the uniform distribution of nanoparticles on hydrogel and their spherical morphology (Figure 9). The rheology test showed the ability of the formulation to coat eye surface and settled in position with eye blinking. The concentration of CBGA uptake into cornea and lens was evaluated versus control after treatment for 4 h. Although only ~0.015% of the CBGA initial load penetrates the cornea, the content of the drug in cornea and lens was four and two times greater than control in the absence of lachrymal drainage. It is expected that the uptake of drugs can be could be even greater in under-eye conditions [105].

## 6. Gene Therapy

Based on the definition, gene therapy can be defined as a technique that allows the clinicians to treat or prevent different types of diseases by delivering the genes, instead of drugs, to the cells [106]. Until this time, it has been shown that gene therapy can be considered as the most promising approach in terms of cornea treatments [107]. In fact, gene therapy possesses the capability of fixing the root of the diseases, in contrast, other strategies may only solve the problem superficially. In case of cornea gene therapy, there is number of characteristic features assigned to the cornea which potentially make it suitable target for gene therapy purposes. The simple, comparatively histological structure, ease of access for both examination and manipulation, its potential to be maintained as ex vivo culture for a relatively long duration, and immune privilege, are some of these features [106,108]. Two viral and non-viral strategies have been employed for gene therapy, but both of them are accompanied by some defects [109]. Although viruses used as viral vector prove satisfactory results, their application needs to be limited due to side effects and toxicity concerns. On the other side, non-viral vectors cannot be a good option for cornea gene therapy either, and this is because of their low transfection efficiency [110]. Among the existing methods, it seems the use of nanoparticule-based corneal gene therapy could be the best. According to their nano-size structure, they can potentially make the intracellular delivery possible [111]. Several studies have tried to demonstrate the capabilities of these nanostructures for cornea gene therapy applications. In one study, Sharma et al. proposed polythymidine conjugated to gold nanoparticles (PEI2-GNPs). The efficiency and toxicity of these nanostructures were evaluated for gene transferring to human cornea in vitro and rabbit cornea in vivo. Their results demonstrated a great transgene delivery without any change of the viability or phenotype of the cells. In addition, based on their observations, no inflammation or redness effects with only moderate cellular death or immune response in rabbit cornea tissue were recorded [110]. PEI-DNA nanoparticles exhibited great potential for transferring growth factor-β type II receptor (sTGFβRII) to control cornea fibrosis in vivo [112]. The effects of bone morphogenetic proteins (BMP7) for gene transfer to inhibit cornea fibrosis was investigated by Tandon et al. [113]. In conclusion, gene therapy with PEI2-GNPs used for BMP7 transfer provided a significant decrease in cornea haze. In continuation of the last research, this time, a combination of two therapeutic genes, BMP7 and hepatocyte growth factor (HGF), for corneal gene therapy was proposed [114]. The purpose behind such a model was to target the initial signal, and the downstream myofibroblast apoptosis which would be obtained by BMP7 and HGF, respectively.

Linear 22-kDa polyethyleneimine (PEI)-DNA nanoparticles were prepared for cornea gene therapy purposes [115]. The effects of PEI nitrogen-to-DNA phosphate (N:P) ratio on gene transfection efficiency were examined. The best ratio for achieving the highest transfection efficiency was N:P 30, with no significant cytotoxicity [115]. Solid based nanoparticle was developed by Torrecilla et al. for short-hairpin RNA (shRNA) gene delivery with the main objective of downregulation metalloproteinase 9 (MMP-9) in cornea cells for neovascularization treatment [116]. Recently, two different types of positive surface charge solid nanoparticles with different ligands (protamine, dextran, or hyaluronic acid (HA)) and synthesized by polyvinyl alcohol (PVA) were developed by Vicente-Pascua et al. [117]. Due to the presence of PVA in their formulation, the cornea penetration was enhanced, and based on their observations, no histological change was found in the cornea tissue. Moreover, among all the prepared solid nanoparticles, the one combined with hyaluronic acid (HA) proved to be the most efficient formulation [117,118].

## 7. Conclusions

Corneal drug delivery is one of the most attractive and challenging activities for researchers because, for example, in 2020, 2.2 billion people suffered from vision impairment or blindness, and for the treatment of most ocular diseases, repeated intraocular injections are necessary. To resolve this problem, a non-invasive drug delivery system could help and decrease the risks of injection into the eyes. The cornea consists of five layers (epithelium, Bowman’s membrane, stroma, Descemet’s membrane, and endothelium) and all of them refract light through the lens on the retina. The structure of the cornea is so unique that the delivery of the drug with drug delivery systems is both difficult and attractive. Although different researches have undertaken in vitro and in vivo studies in this area during the last decades, so as to find the best method to deliver drugs to the cornea, researchers need more details before using the ideal system in a clinical context [1].

Even though different methods, such as topical methods, eye drops, ointments, contact lenses, punctual plugs, etc., have been used for corneal drug delivery, these methods increase the risk of the blindness and are not very useful for all corneal problems. Consequently, a corneal drug delivery system can help people all around the world to have more effective therapeutic alternative to conventional therapies for cornea damages and problems [48]. The main properties of the cornea drug delivery are the ability to control the release rate of the drugs and enhance drug bioavailability. With this useful method, drugs can be delivered to specific parts of the cornea, with a controllable release rate, thereby decreasing the amount of precorneal drug loss and increasing drug bioavailability. Additionally, with these smart methods, a drug could remain in prolonged contact with different parts of the cornea [49].

Ocular drug delivery systems are so challenging because of the anatomy of this tissue, critical and specific environment that exists in this important tissue. During the last decades, to overcome the limitations of traditional methods, a considerable amount of research has been undertaken to improve the efficiency of drug delivery systems, and each of them has some advantages and disadvantages. One of the most important materials that researchers are focused on is nanoparticles. Nanostructure materials are able to encapsulate and deliver small molecules so as to deliver them to various parts of the cornea. In scientific societies, materials with a size between 1 to 1000 nm in at least one dimension are called nanoparticles and it must be noted that using larger size particles (more than 1000 nm) for ocular drug delivery has a negative effect on the treatment and causes ocular irritation [47]. Researchers have tried to fabricate and use various kinds of nanoparticles, like liposomes, dendrimers, polymeric nanoparticles, niosomes, microemulsions, nanosuspensions, and hydrogels for corneal drug delivery with the purpose of encapsulating, carrying, and delivering drugs, proteins, and peptides to specific parts of the cornea in order to enhance efficiency by improving drug penetration and reduce side effects because of necessary lower dosage requirements [46]. To improve the efficiency of the drugs and reduce the amount of precorneal drug loss, nanoparticles have to rapidly penetrate the corneal membrane and deliver especial drugs or small molecular to the specific part. Consequently, the drug could remain in prolonged contact with different parts of the cornea [45].

Different key approaches are suggested to improve drug delivery systems and enhance the efficiency of therapy. However, the best way for increasing efficiency and improving drug penetration is using a biodegradable nanocarrier for corneal drug delivery. These kinds of materials have shown a good ability for corneal drug delivery systems and by using these kinds of nanoparticles, the residence time of the drug was enhanced in various parts of the cornea, the drug can release sustainably, and the nanocarrier degrades after a specific time, etc. [57]. Additionally, these kinds of materials will degrade after a specific time, and importantly, this property will help patients to reduce the number of visits made to the doctor. A variety of nanocarriers, such as liposomes, dendrimers, polymeric nanoparticles, niosomes, microemulsions, nanosuspensions, and hydrogels, have been developed for ocular drug delivery. All of these biodegradable nanoparticles have some positive and negative aspects, and each of them is useful for a specific layer of the cornea and a particular drug. Additionally, by utilizing biodegradable nanoparticles, genes can be delivered to the cells instead of drugs. It should be mentioned that this method is the most promising approach in terms of corneal treatments [113].

All in all, biodegradable nanoparticle formulations with the help of different polymers offer us a new strategy to help to release drugs sustainable at a particular cornea layer and consequently, the bioavailability of drugs can be increased. Even though many have been researching biodegradable nanoparticles for corneal wound healing during the last decades, more clinical studies are essential to offer additional information and insights related to these kinds of great biomaterials in corneal drug delivery [119].

## Figures and Tables

**Figure 1 pharmaceutics-12-01232-f001:**
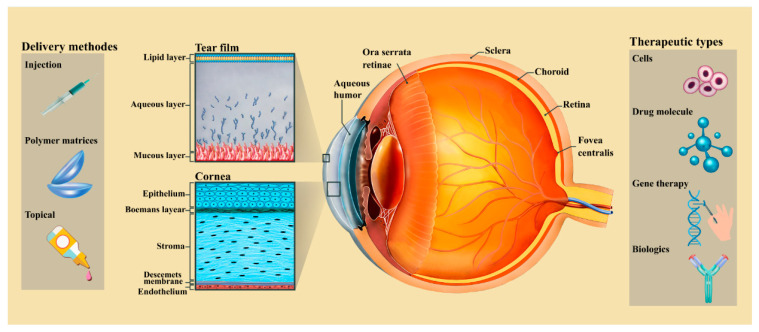
Schematic images of five layer of human cornea structure and overview of categories of cornea drug delivery systems that are adopted to a wide variety of therapeutic agents and their related permeability barriers. These methods include topical administration, drug loaded ocular inserts (for either punctal placement, conjunctival or cornea), and injections through the cornea [1].

**Figure 2 pharmaceutics-12-01232-f002:**
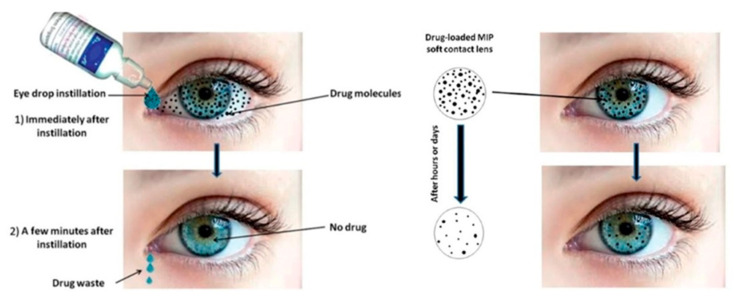
The superiority of soft drug loaded contact lenses in comparison to eye drops. Adapted with permission from [12], J. Pharmacol. Exp. Ther., 2019.

**Figure 3 pharmaceutics-12-01232-f003:**
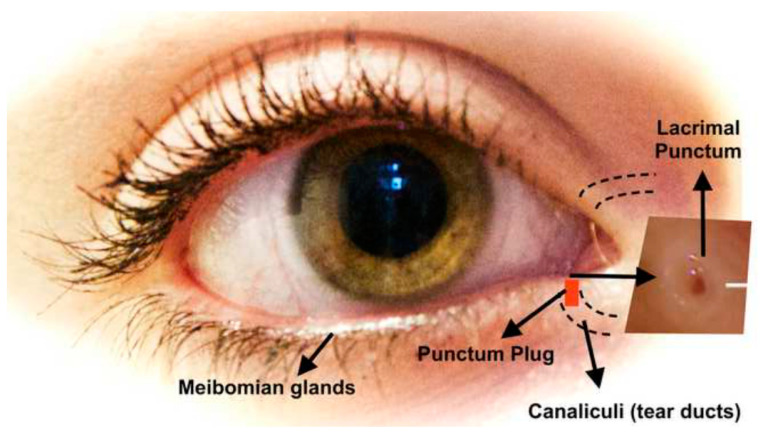
A view of punctum location and position of it in the eye. Adapted with permission from [36], Drug Discov. Today, 2015.

**Figure 4 pharmaceutics-12-01232-f004:**
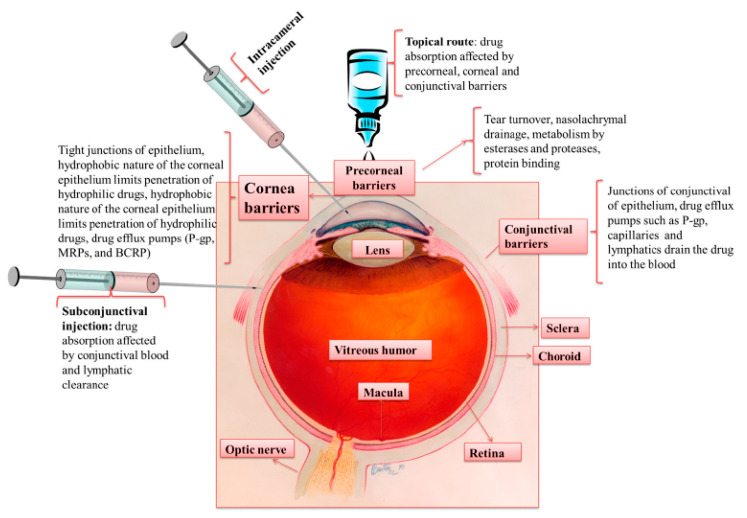
Routes of drug administration for anterior segment of the eye, clearance mechanisms and ocular tissue barriers that avoid drug absorption into the eye [40].

**Figure 5 pharmaceutics-12-01232-f005:**
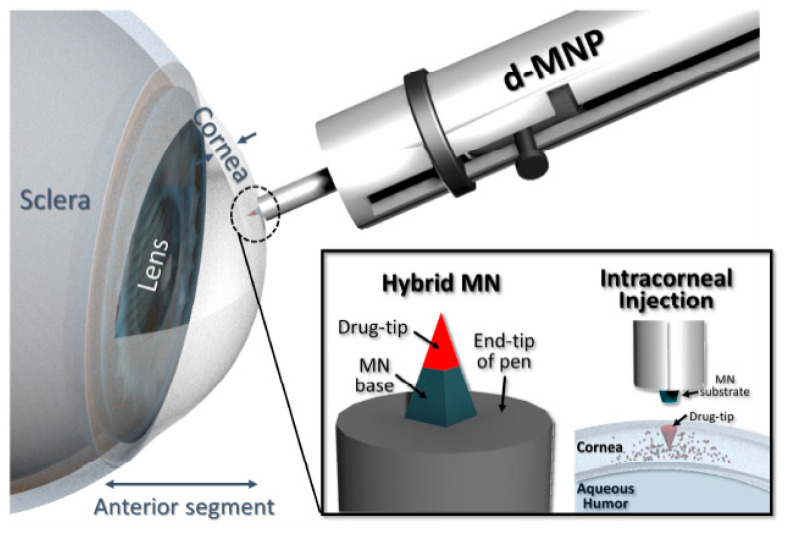
An overview of sustained drug release by microneedle. Adapted with permission from [45], Acta Biomater., 2018.

**Figure 6 pharmaceutics-12-01232-f006:**
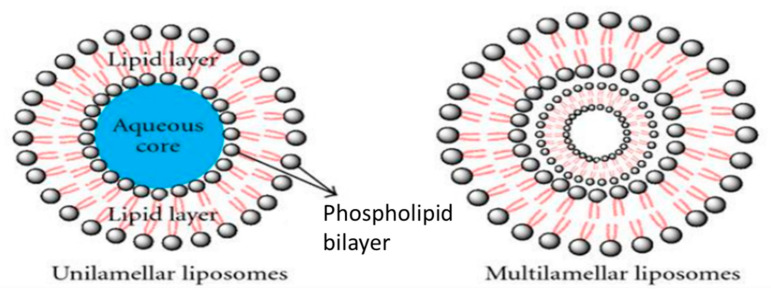
Schematic representation of Liposomes. Adapted from [69], Journal of Drug Delivery, 2011.

**Figure 7 pharmaceutics-12-01232-f007:**
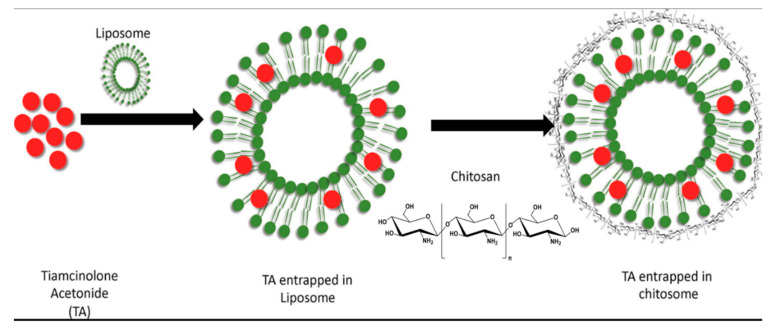
Schematic of chitosan coated liposomes. Adapted with permission from [76], Int. J. Biol. Macromol. 2020.

**Figure 8 pharmaceutics-12-01232-f008:**
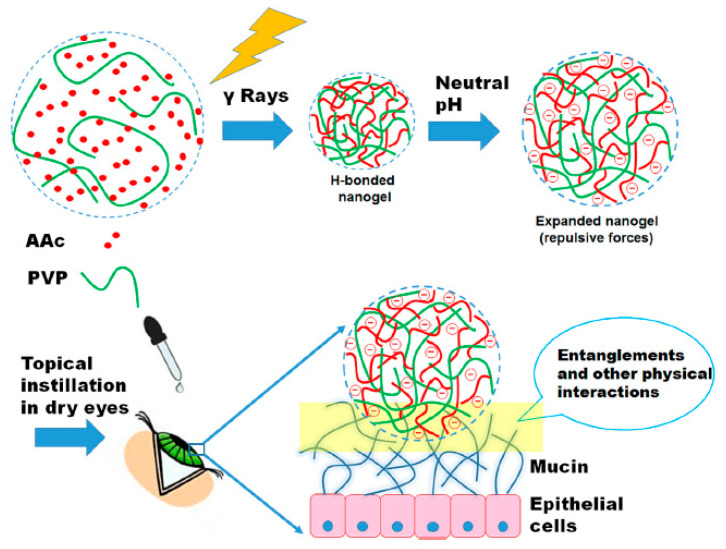
An overview of the preparation mechanism of PVP/PAAc nanogel [49].

**Figure 9 pharmaceutics-12-01232-f009:**
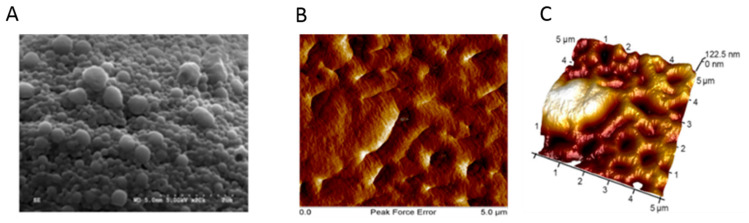
SEM and AFM analysis of the NPs and composite. (**A**) SEM analysis shows spherical morphology of NPs with an average diameter of 186 nm. (**B**,**C**) AFM result exhibits uniformly distribution of NPs embedded in hydrogel and formed a soft surface. Adapted from [105], Drug Deliv. Transl. Res. 2018.

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
