# Peer review of "Biodegradable Nanoparticle for Cornea Drug Delivery: Focus Review"

_pharmaceutics, 2020, doi:10.3390/pharmaceutics12121232_

Round 1
Reviewer 1 Report
In this manuscript “Biodegradable Nanoparticle for Corneal Drug Delivery: Focus Review” Raahemifar and co-workers review comprehensively the strategies for drug delivery to corneal structures using soft nanoplatforms: liposomes, dendrimers, polymeric nanoparticles, niosomes, microemulsions, nanosuspensions and hydrogels.
I found the review difficult to read due to a “deficient” domain of the English language and grammar and at times scientific imprecisions.
The review seems too long and “wordy” with some repetition of ideas (and even phrases).
Still, I think that the subject and scope of the review is suitable for this Journal and likely to appeal to a wide audience of researchers interested in nanotechnology for drug delivery applications in general and to the eye in particular.
I believe that this review can be published after major revisions.
The authors should consider summarizing and systematizing the information in table format making evident the “advantages” and “disadvantages” of the different drug micro-nanocarriers in relation to the targeted corneal structures and the delivered drugs.
Author Response
Dear reviewer,
I hope you and your family are safe in this situation. I have correct all point that you mentioned.
Best.
Reviewer 2 Report
1-During last decades, researchers all around the world has focused on the characteristic pros and cons of the different drug delivery systems for this important tissue change it for sense organ.
2-Nanoparticles are being investigated with the purpose of improving drug delivery systems for corneal. Corneal diseases
3-Biodegradable nanocarrier for corneal drug delivery is one of the most useful method for corneal drug delivery. Drug delivery repeated.
4-The aggressive treatments are frozen therapy, laser therapy, change it for frozen or lases therapy..
5-The normal cornea is a dome-shaped, avascular, a transparent connective tissue
6-involving of five layers
7-rewrite: and all of them refract light through the lens on the retina but because of the different anatomical and physiological berries, the efficient of the drugs is limited. Various factors limiting the bioavailability of ocular drugs for patients like tears, conjunctiva, sclera, blood-ocular barriers, and corneal structure.
8-To overcome this problem, different drug delivery systems have improved so as to increase the effectiveness of drugs for ocular [45].
9-On the one hand, various methods such as topical methods, eye drops
Change it for like eye drops….
10-a microemulsion is not nanoparyicle….this is a mistake and check which ones are nanoparticles..of them
Rewrite Researchers have been tried to fabricate various kinds of nanoparticles like liposomes, dendrimers, polymeric nanoparticles, niosomes, microemulsions, nanosuspensions and hydrogels for corneal drug delivery with the purpose of encapsulating,
11-references in
The aggressive treatments are frozen therapy, laser therapy, surgery, and drug administration by periocular or intraocular injection and these methods do not have decent effects on some patients and have high recurrence rates [1]. The normal cornea is a dome-shaped, avascular, a transparent connective tissue, and unique in a structure that covers the front portion of the eye and this precise tissue is one of the most densely innervated tissues in the body because it exposed to the external environment. This important tissue involving of five layers (epithelium, Bowman’s membrane, stroma, Descemet’s membrane, and endothelium), and all of them refract light through the lens on the retina but because of the different anatomical and physiological berries, the efficient of the drugs is limited. Various factors limiting the bioavailability of ocular drugs for patients like tears, conjunctiva, sclera, blood-ocular barriers, and corneal structure. To overcome this problem, different drug delivery systems have improved so as to increase the effectiveness of drugs for ocular [45]. On the one hand, various methods such as topical methods, eye drops, ointments, contact lenses, punctual plugs, and have been using for corneal drug delivery. On the other hand, nanostructures materials that are able to encapsulate and deliver small molecules so as to deliver them to various parts of the corneal. For having better efficiency of the drugs and reduce the amount of precorneal drug loss, nanoparticles have to rapidly penetrate the corneal membrane and deliver especial drugs or small molecular to the specific part. Consequently, the drug could remain in prolonged contact with different parts of the corneal [47].
1, 45, 47?????? Where are 2,3,4….etc they are not in order…
12-this sentence is not clear, rewrite
On the other hand, the stroma layer be made of extracellular matrix (ECM), keratocytes, and glycosaminoglycans
13-corneal is an adjective and has been confused in all the manuscript. It should appear cornea when it is referred to the organ.
14-For anterior chamber diseases, in former times, ophthalmic drug delivery was used by topical formulations ointments, emulsions, gels, solutions, and suspensions. Change it for ..or supensions.
15-change it for
As expected, after topical drug instillation, the concentration of the drug is high and after that concentration quickly declines. Decrease
16-Ocular ointments as the carrier systems developed for the topical administration of drugs,
17.It would be very convenient to insert a new figure describing all the sections
18.
Rewrite
The eye drops contain lower than 15% cyclodextrins concentration was attained optimal bioavailability [23]
19-
In 3.1.3. Ocular gel The achievement of an optimal drug concentration at the target area is the main problem with topical ocular therapeutics. It sounds repeated …….all sections begin the same.
20-The first marketed device was the pilocarpine Ocusert® (ALZA Corp.) that reach zero-order kinetics. Explain ths in detail.
21-too many repeated punctum
.. Figure 3. A view of punctum location and position of punctal plugs placed in the punctum of the eye [36].
22-too big
Figure 5. An overview of sustained drug release utilizing a detachable hybrid microneedle [45].
23-4. Nanostructured materials and properties
Insert other uses of nanomaterials in biomedicine, examples in the market
24-decent is not formal English
For having decent drug
25-By using nanoparticles as nanocarrier, it is possible to deliver drugs or small molecular to the posterior and anterior segment ocular [55].
Small molecules or entities
26- in
Figure 6. schematic representation of nanocarriers using for ocular drug delivery: (a) Liposomes, (b) Dendrimers, (c) Niosomes [59], (d)Polymeric nanoparticles [60], (e) Microemulsions, (f) Nanosuspensions [61], (g) Hydrogels [62].
Only nanohydrogels are nanocarriers, the same for microemulsions-… revise this deeply.
27-in Figure 7. schematic representation of Liposomes and schematic of liposomes preparation procedure [74].
Phospholipid is wrong written
28-
In fig 8
Figure 8. schematic of chitosan coated liposomes [80]. Chitosan structure should be bigger.
29-in fig
Figure 9. Schematic representation of dendrimers and schematic of Niosomes [83].
More resolution of images are needed
30-fig 12 and 13 have not enough resolution, are the copied from other journal, what about copy right?
31-
oligonucleotides due to the mentioned forming process and existence of an aqueous internal environment Hence,
hence
32-Figure 14. An overview of the prepration mechanism of PVP/PAAc nanogel (49).
Preparation
33-It should be included a section concerning prospective studies and simulation studies concerning these drugs
Author Response
Dear reviewer,
I hope you and your family are safe in this situation. First, I have to say thank you for your outstanding comments and I have correct all of them to improve our article.
Best.
Round 2
Reviewer 2 Report
The comments have been added to the manuscript as recommended.